# Diverse mutant selection windows shape spatial heterogeneity in evolving populations

Eshan S. King[1], Dagim S. Tadele[2,3], Beck Pierce[4], Michael Hinczewski[5], Jacob G. Scott[1,2,5,6]*

**1** Case Western Reserve University School of Medicine, Cleveland, Ohio, United States of America, **2** Department of Translational Hematology and Oncology Research, Cleveland Clinic, Cleveland, Ohio, United States of America, **3** Oslo University Hospital, Ullevål, Department of Medical Genetics, Oslo, Norway, **4** Department of Mathematics, Applied Mathematics, and Statistics, Case Western Reserve University, Cleveland, Ohio, United States of America, **5** Department of Physics, Case Western Reserve University, Cleveland, Ohio, United States of America, **6** Department of Radiation Oncology, Cleveland Clinic, Cleveland, Ohio, United States of America

* scottj10@ccf.org

**Funding:** JGS and ESK were supported by NIH 5R37CA244613-04 (https://www.cancer.gov/). ESK was supported by NIH 3T32GM007250-46S1 (https://www.nigms.nih.gov/). JGS was supported

## Abstract

Mutant selection windows (MSWs), the range of drug concentrations that select for drug-resistant mutants, have long been used as a model for predicting drug resistance and designing optimal dosing strategies in infectious disease. The canonical MSW model offers comparisons between two subtypes at a time: drug-sensitive and drug-resistant. In contrast, the fitness landscape model with $N$ alleles, which maps genotype to fitness, allows comparisons between $N$ genotypes simultaneously, but does not encode continuous drug response data. In clinical settings, there may be a wide range of drug concentrations selecting for a variety of genotypes in both cancer and infectious diseases. Therefore, there is a need for a more robust model of the pathogen response to therapy to predict resistance and design new therapeutic approaches. Fitness seascapes, which model genotype-by-environment interactions, permit multiple MSW comparisons simultaneously by encoding genotype-specific dose-response data. By comparing dose-response curves, one can visualize the range of drug concentrations where one genotype is selected over another. In this work, we show how $N$-allele fitness seascapes allow for $N * 2^{N-1}$ unique MSW comparisons. In spatial drug diffusion models, we demonstrate how fitness seascapes reveal spatially heterogeneous MSWs, extending the MSW model to more fully reflect the selection of drug resistant genotypes. Furthermore, using synthetic data and empirical dose-response data in cancer, we find that the spatial structure of MSWs shapes the evolution of drug resistance in an agent-based model. By simulating a tumor treated with cyclic drug therapy, we find that mutant selection windows introduced by drug diffusion promote the proliferation of drug resistant cells. Our work highlights the importance and utility of considering dose-dependent fitness seascapes in evolutionary medicine.

by American Cancer Society Research Scholar Grant RSG-20-096-01 (https://www.cancer.org/). DST was supported by The Research Council of Norway with grant 325628/IAR (https://www.forskningsradet.no/en/). The funders had no role in study design, data collection and analysis, decision to publish, or preparation of the manuscript.

**Competing interests:** The authors have declared that no competing interests exist.

## Author summary

Drug resistance in infectious disease and cancer is a major driver of mortality. While undergoing treatment, the population of cells in a tumor or infection may evolve the ability to grow despite the use of previously effective drugs. Researchers hypothesize that the spatial organization of these disease populations may contribute to drug resistance. In this work, we analyze how spatial gradients of drug concentration impact the evolution of drug resistance. We consider a decades-old model called the mutant selection window (MSW), which describes the drug concentration range that selects for drug-resistant cells. We show how extending this model with continuous dose-response data, which describes how different types of cells respond to drug, improves the ability of MSWs to predict evolution. This work helps us understand how the spatial organization of cells, such as the organization of blood vessels within a tumor, may promote drug resistance. In the future, we may use these methods to optimize drug dosing to prevent resistance or leverage known vulnerabilities of drug-resistant cells.

## Introduction

Drug resistance in cancer and infectious disease is governed by the unifying principles of evolution. Selection, which is integral to evolution, may be described by dose-response curves, which model growth rate as a function of drug concentration. Genotype-specific dose-response curves are ubiquitous across disease domains, including cancer and infectious disease. Dose-response curves can vary between different genotypes in multiple characteristics, such as their y-intercept (drug-free growth rate), $IC_{50}$ (half-maximal inhibitory concentration), and shape [1–9]. Dose-response curves may also reveal fitness tradeoffs, or costs, where drug resistance imposes a fitness cost in the drug-free environment [10–12]. These diverse collections of dose-response curves among individual disease states give rise to varying degrees of selection when the drug concentration varies in time and space. Collections of genotype-specific dose response curves constitute fitness seascapes, which extend the fitness landscape model by mapping both genotype and environment (i.e. drug concentration) to fitness (Box 1 and Fig 1A) [1–3, 13–16].

The concept of the mutant selection window (MSW), the range of drug concentrations where the mutant growth rate is not fully suppressed and exceeds the wild-type growth rate, has been studied as a means of predicting evolution and optimizing drug regimens [1, 5, 6, 17–22]. Under the MSW paradigm, drug regimens should be chosen that minimize the time a patient is subject to a drug concentration within the MSW (Box 1 and Fig 1B) [22, 24]. Das et al. have previously demonstrated that MSWs are intrinsically embedded in fitness seascapes [1]; by comparing dose-response curves in a fitness seascape, one can visualize the range of drug concentrations that selects for a drug resistant mutant. Previous work has shown how patient nonadherence and drug gradients caused by tissue compartmentalization confound the use of MSWs in optimizing drug regimens, allowing for the emergence of drug resistant mutants [5, 6, 25, 26]. Incorporating PK/PD models and MSWs may be crucial for translating evolutionary medicine to the clinic– for instance, Pan et al have demonstrated the presence of an MSW for *Pseudomonas aeruginosa* in an *in vivo* model, revealing a correlation between sub-optimal PK/PD parameters and the emergence of drug resistance [27]. In addition, drug diffusion from blood vessels has been found to be a driver of drug concentration gradients in cancer, potentially limiting the effectiveness of chemotherapies [28–30].

## Box 1: Fitness seascape and MSW terminology

### Fitness seascapes

Fitness seascapes are extensions of the fitness landscape model that map both environment and genotype to fitness. For instance, while a fitness landscape may map genotype to minimum inhibitory concentration (MIC), a measure of drug resistance, a fitness seascape may map genotype and drug concentration to growth rate. Here, we model fitness seascapes as collections of genotype-specific dose-response curves. Previous usage of the term 'fitness seascape' has referred specifically to time-varying fitness landscapes. However, implicit in this use is that the environment shapes the fitness landscape, and it is the time-varying dynamics of the environment that result in the time-varying fitness landscape. In this work, we propose expanding the definition of fitness seascape to include any mapping from genotype and environment to fitness. Formally, we can define the relationship between fitness seascapes and fitness landscapes as

$$S([c]) = \mathcal{L}_{[c]}, \tag{1}$$

where $S([c])$ represents the fitness seascape as a function drug concentration and $\mathcal{L}_{[c]}$ represents the fitness landscape at a given drug concentration. In Fig 1A, genotypes are modeled as binary strings of length 2, where each position in the string indicates the presence or absence of a particular point mutation. Each genotype is associated with a corresponding dose-response curve. Dose-response curves are modeled by:

$$g([c]) = \frac{g_{drugless}}{1 + e^{(IC_{50}-[c])/v}}, \tag{2}$$

where $g_{drugless}$ is the genotype-specific growth rate in the absence of drug, $IC_{50}$ is the half-maximal inhibitory concentration, and $v$ is the Hill coefficient, which determines the steepness of the curve. The collection of genotype-specific dose response curves constitutes a fitness seascape, where fitness is a function of both genotype and drug concentration. Rank-order fitness landscapes that describe the relative fitness rank of each genotype at $10^{-2}$ and $10^2$ μg/mL drug are shown inset in Fig 1A. The fitness landscape changes as a function of drug concentration due to the dose-response curves associated with each genotype. Fitness costs to drug resistance are also intrinsically embedded in fitness seascapes. The wild-type 00 genotype exhibits the highest growth rate in the absence of drug, while genotype 11, which has the highest $IC_{50}$, exhibits the lowest growth rate in the absence of drug.

### Mutant selection windows

Mutant selection window refers to the range of antibiotic concentrations that select for a drug resistance mutant without fully suppressing growth [1, 5, 6, 17–22]. MSWs are important to consider because they are thought to facilitate the evolution of antibiotic resistance. MSWs have been used to design dosing regimens that minimize the time that a patient's serum drug concentration is within the MSW ($t_{MSW}$) [21, 23]. Fig 1B shows a simulated patient's serum drug concentration following a single antibiotic dose. The MSW for this particular microbe is 0.35–0.65 in the normalized range. The time spent within the MSW, $t_{MSW}$, is indicated by the blue shaded region. While the patient's serum concentration is within this range, the drug resistant mutant is selected for without being fully suppressed.

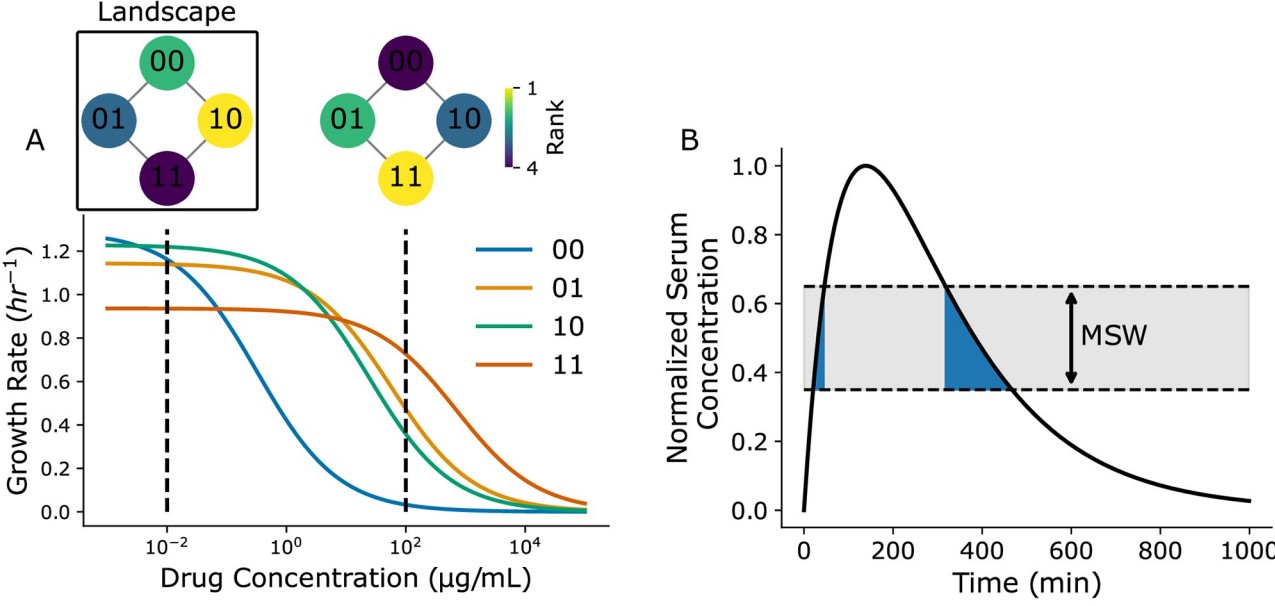

**Fig 1. Illustrations of fitness seascapes and mutant selection windows. (A)** Example fitness seascape parameterized with random dose-response data. Genotypes are modeled with binary strings, where '0' represent the absence of a point mutation at a specific position and '1' represents the presence of said mutation. Corresponding rank-order fitness landscapes are annotated at $10^{-2}$ and $10^2$ μg/mL. **(B)** Example patient serum drug concentration profile over time for a single drug dose. The mutant selection window range is annotated in gray, while the time the serum drug concentration resides within the mutant selection window, $t_{MSW}$, is shown in blue.

MSWs traditionally offer comparisons between two genotypes or phenotypes at a time—i.e., between a drug susceptible reference genotype and a drug resistant genotype. By examining MSWs with the fitness seascape model, we may compare many genotypes simultaneously. For instance, in the binary landcsape model with *N* mutational sites, each genotype may be simultaneously compared to *N* adjacent genotypes (genotypes that differ by 1 mutation) [3, 31–37]. Here, we expand on this idea and explore the MSW model through the lens of fitness seascapes. First, we illustrate how an *N*-allele fitness seascape allows for $N * 2^N$ MSW comparisons at a time. Then, we derive the steady-state drug concentration profile for drug diffusion from blood vessels in 1 and 2 dimensions, revealing the presence of heterogeneous MSWs. Using a 2-D agent based model, we explore how drug diffusion shapes MSW spatial heterogeneity and how MSW spatial structure impacts evolution. By parameterizing our model with novel empirical dose-response data in non-small cell lung cancer, we simulate a tumor treated with cyclic drug therapy. We find that mutant selection windows driven by drug diffusion promote drug resistance and result in treatment failure. While previous work has analyzed the importance of time spent in a MSW, we also consider the impact of space occupied by a MSW. We argue that both time and space occupied by a MSW may impact the emergence of resistance.

This work further explores the connection between mutant selection windows and fitness seascapes using realistic pharmacokinetic models. Our results highlight the utility of fitness seascapes in modeling evolution when drug concentration varies in space. Furthermore, because of the multiplicity of MSWs present in a fitness seascape, this work suggests that a higher-dimensional MSW model offered by fitness seascapes may be more powerful for predicting evolution in clinical settings, particularly when concerned with population heterogeneity.

## Results

### Fitness seascapes embed mutant selection window data

We first investigated how multiple mutant selection windows are embedded in fitness seascapes, as demonstrated by Das et al. [1]. Here, we model genotypes as binary strings of length $N$, with a zero indicating no mutation and a 1 indicating a mutation in a given allelic position. $N$ represents the total number of resistance-conferring mutations considered. These mutations can be thought of as either single nucleotide polymorphisms, single amino acid substitutions, or larger chromosomal changes. To illustrate this, we model a simple 2-allele fitness seascape by mapping a combinatorially complete set of genotypes (00, 01, 10, 11) to independent, randomly generated dose-response curves (Fig 1A). Dose response curves differ in their $IC_{50}$ and drug-free growth rates. Empirical fitness seascapes bearing a similar structure and demonstrating fitness tradeoffs have been reported by others across different kingdoms of life, providing a theoretical foundation for this approach [1–3].

In an $N$-allele model, each genotype can be compared to $N$ neighboring genotypes in the landscape (Fig 1A). Here, 'neighboring' refers to genotypes that differ by one genetic change, or Hamming distance 1 (i.e., 00 and 01 are neighbors, but not 00 and 11). When calculating mutant selection windows, one first defines a wild type or 'reference genotype' to compare to the mutant. Given that each genotype in a fitness seascape can itself be thought of as the reference genotype and compared to its $N$ neighbors, an $N$-allele fitness seascape embeds $N * 2^N$ MSW comparisons. If one includes only *unique* MSW comparisons, i.e. the distinction between reference and comparison genotype is not meaningful, then this expression is $N * 2^{N-1}$. Fig 2B shows the grid of all possible MSWs for a 2-allele seascape, shaded by the selection coefficient $s_{i,j}$. The selection coefficient is defined as $s_{i,j} = \frac{g_i}{g_j}$, where $g_i$ represents the growth rate of the more fit genotype and $g_j$ represents the growth rate of the less fit genotype. Reference selection refers to the range of drug concentrations where the reference genotype has a higher fitness than the mutant. Similarly, mutant selection refers to the range of drug concentration where the mutant has a higher fitness. Net loss refers to the range of drug concentrations where the net replication rate of the reference and mutant genotypes are both less than 0. An example of how MSWs are calculated with dose-response curves is shown in Fig 2A. Notably, the strength of selection blurs between the boundaries of the selection windows, further complicating the MSW paradigm (Fig 2B). It may be useful to consider where selection is strongest within a MSW when designing dosing strategies.

### Multiple mutant selection windows arise when drug concentration varies in space

We next sought to investigate MSWs in physiologically-relevant spatial models of drug drug diffusion in tissue. For the 1-dimensional (1D) case, we consider a location $x$ at a time $t$. The drug concentration $u(x, t)$ arising from a blood vessel source can be modeled with drug diffusion rate $D$, drug clearance rate $\gamma$, and the drug concentration at the tissue-blood vessel boundary $k$. We use a partial differential equation (Eq (3)) to find a steady state solution of drug diffusion:

$$\frac{\partial}{\partial t} u(x, t) = D \frac{\partial^2}{\partial x^2} u(x, t) - \gamma u(x, t) + k\delta(x), \tag{3}$$

where $\delta(x)$ is the Kronecker delta function representing a blood vessel modeled as a point

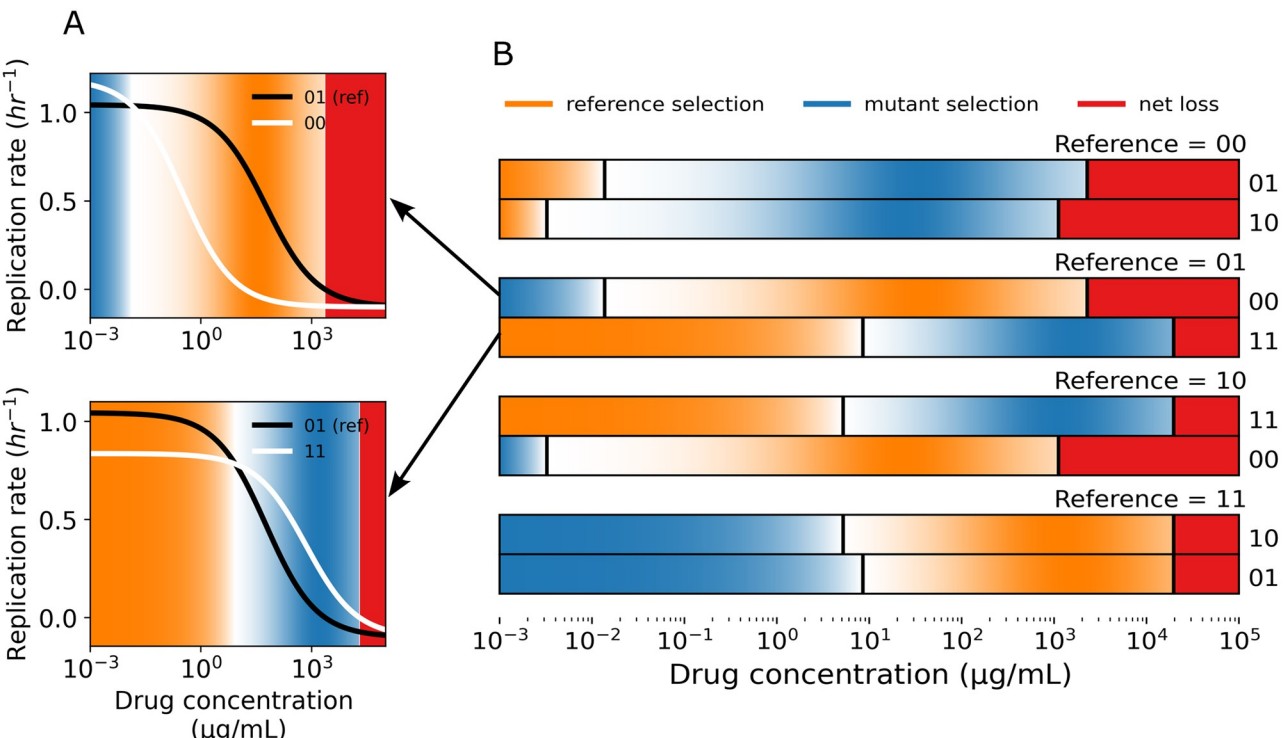

**Fig 2. Fitness seascapes embed mutant selection window data. (A)** Example MSWs shown with the corresponding dose-response curves. The black line (01) is the dose-response curve for the reference genotype, while the white lines are considered the mutant genotypes. Orange corresponds to the reference selection window, blue to the mutant selection window, and red corresponds to the drug concentration that inhibits growth of both genotypes (net loss). **(B)** All 8 (8 = $N * 2^N$, $N$ = 2) MSW comparisons for a 2-allele fitness seascape. Each two row group represents a single reference genotype compared to its two neighboring genotypes. Reference selection, mutant selection, and net loss windows are calculated as a function of drug concentration and shown as colored rectangles. Selection windows are shaded by the normalized selection coefficient, $s_{i,j}$, which is defined as the normalized ratio between the growth rates under comparison, $s_{i,j} = \frac{g_i}{g_j}$, where $g_i$ represents the growth rate of the genotype under selection and $g_j$ represents the growth rate of the less fit genotype. All dose-response data is taken from the fitness seascape in Fig 1A.

source. The 1D steady-state solution is of the form:

$$u(x) = \frac{k}{\sqrt{4D\gamma}}\exp\left(-|x|\sqrt{\frac{\gamma}{D}}\right).$$  (4)

The clearance rate $\gamma$ summarizes drug metabolism, consumption, and clearance into one parameter. We model a blood vessel source at $x = 0$ and compute drug concentration as a function of distance from the blood vessel. Using a similar approach to the time-varying case above, we used information provided by the simulated $N = 2$ fitness seascape to identify MSWs in space. Drug diffusion results in four different MSWs across the simulated 1D tissue patch (Fig 3A). These results demonstrate that, given a constant supply of a drug source from a vessel in a tissue compartment, multiple mutant selection windows may exist simultaneously.

To better understand spatial heterogeneity, we extended the model to 2 dimensions (2D) with a similar partial differential equation as before (see Methods for derivation). The steady-state drug concentration is given by:

$$u(r) = \frac{k}{2\pi D}K_0\left(\sqrt{\frac{\gamma}{D}}r\right),$$  (5)

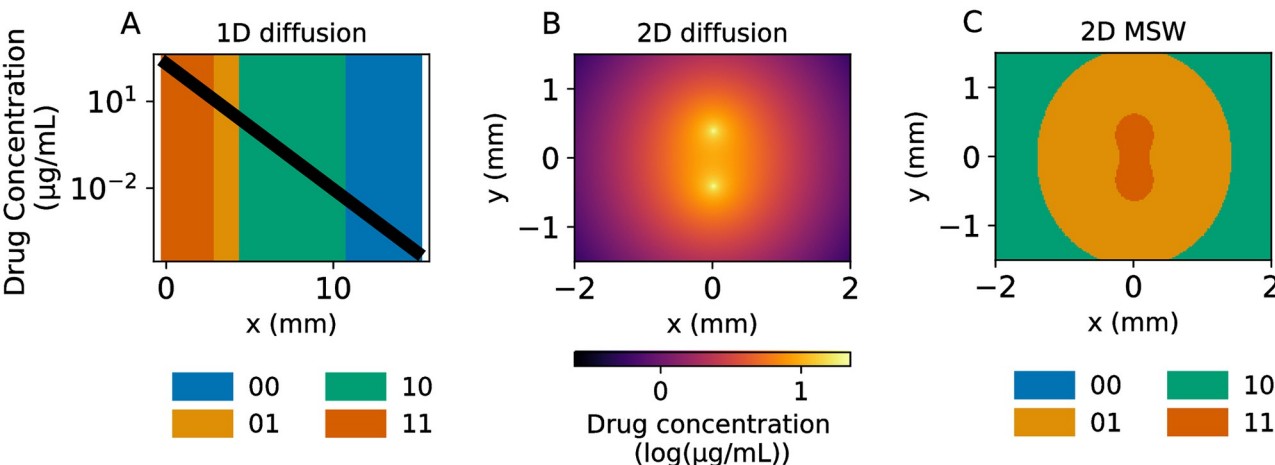

**Fig 3. 1- and 2-D diffusion models reveal diverse MSWs.** (**A**) Simulated 1-D drug diffusion in tissue from a blood vessel placed at x = 0. The black line represents the drug concentration as a function of distance *x* from the blood vessel, while the color represents the MSW at that region. (**B**) 2-D steady state drug diffusion from two blood vessels placed at (0, -0.4) and (0, 0.4). (**C**) Spatial MSWs corresponding to the steady-state diffusion in **B**. All dose-response data is based on the fitness seascape described in Fig 1.

where $K_0(\cdot)$ is a modified Bessel function of the second kind and *r* is the radial distance from the blood vessel source. We then considered the spatial effects of multiple blood vessels as drug sources in two dimensions, allowing for a more sophisticated representation of MSWs in tissue. For convenience, we set $\sqrt{\frac{\gamma}{D}} = 1\text{mm}^{-1}$. The distinction between 1 and 2 spatial dimensions is important, as the space occupied by a MSW in 2D scales by the distance from the blood vessel squared, versus linearly in 1D. The steady-state drug concentration profile is shown in Fig 3B, with two blood vessels placed at (0,-0.4) and (0,0.4). Identifying the MSWs as before reveals 3 distinct windows in this regime (Fig 3C), with the 11 mutant window appearing to 'bridge' between the two blood vessels.

This 2D model offers a more complete picture of the MSWs functionally existing in an area around two blood vessels. These results suggest that including drug diffusion and pharmacodynamic effects may be important for simulating and predicting the evolution of drug resistance spatially. Introducing more blood vessels with arbitrary patterns would further complicate the drug diffusion pattern and the resulting MSWs. Together, these results complicate the notion of a single MSW driving the evolution of drug resistance. Instead, multiple MSWs may dictate evolution within a single population across time and space.

## Drug pharmacokinetics drive spatial heterogeneity

We next sought to understand how the spatial heterogeneity of MSWs may impact genetic heterogeneity in an evolving population. Using the Hybrid Automata Library (HAL) [38], we implemented spatial agent-based simulations parameterized with the synthetic fitness seascape shown in Fig 1. In this work, we use the terms "agents" and "cells" interchangeably. We simulated evolution on a 100-by-100 grid with drug diffusion from two blood vessels placed at $x = 50, y = 25$ and $x = 50, y = 75$ (aligned vertically at the midline of the grid). Each simulation was initiated as a circle with radius 10 and an initial proportion of mutants of 0.01, meaning that each initial cell had a 0.01 probability of having a non-wild-type genotype (i.e. 01, 10, or 11). In addition to beginning the simulation with a heterogeneous population, mutants can arise during the simulation by random mutations during cell division. We also used HAL to

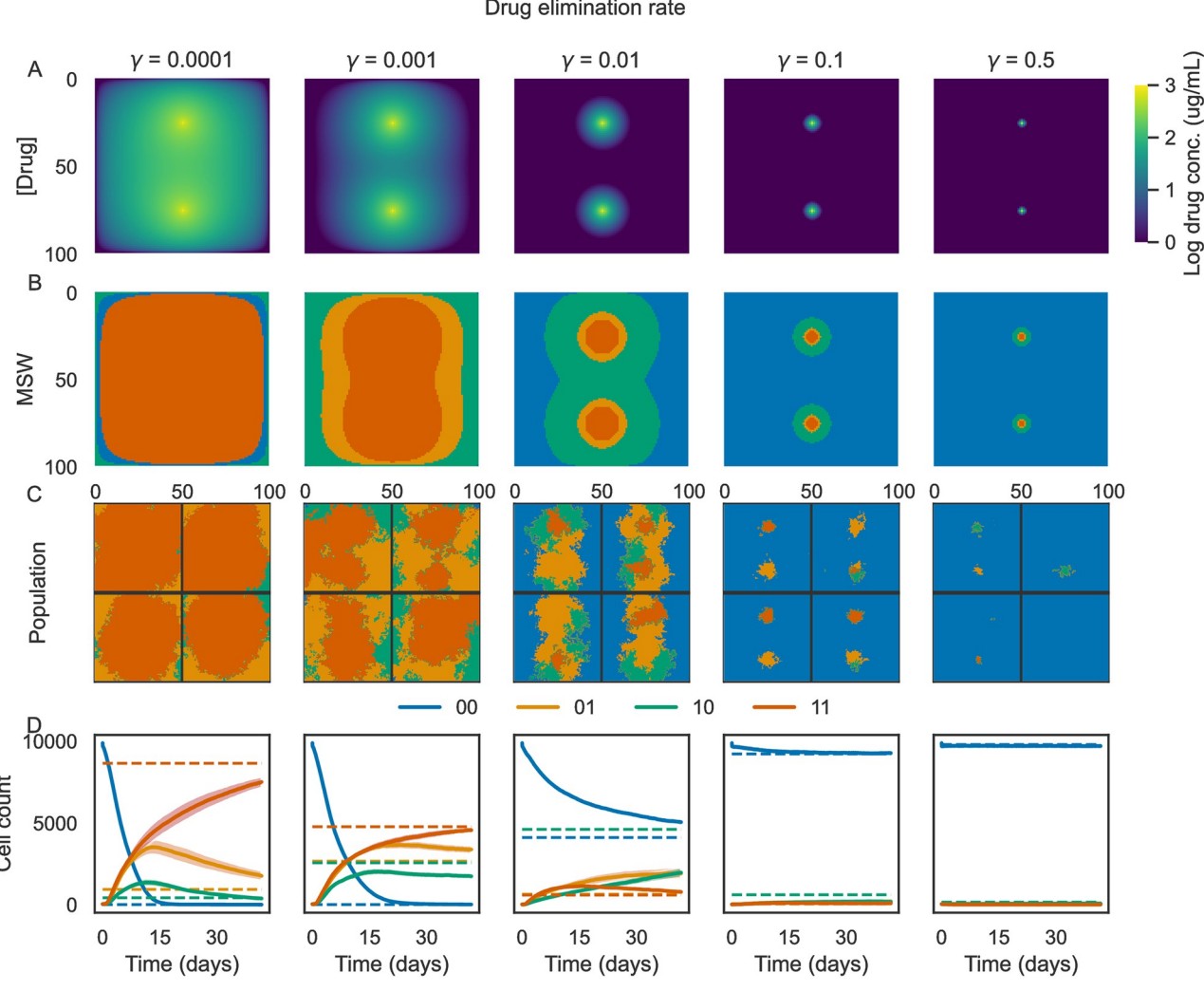

**Fig 4. Varying drug elimination rate promotes MSW and population heterogeneity.** Example results of simulations of cells evolving in a drug concentration gradient from two blood vessels. Each column corresponds to a different drug elimination rate (0 to 0.5, labeled at the top of each column). **(A)** Final drug concentration profile at the end of the simulations. **(B)** Mutant selection windows for the final drug concentration profiles. **(C)** Example simulations. Each quadrant represents an individual simulation. Colors correspond to the cell genotype at each grid position. **(D)** Averaged population counts from simulations for each condition. Colored lines correspond to the average number of cells of each genotype over time, while shading corresponds to the standard error estimate over time ($N = 10$ simulations per condition). Dotted horizontal lines indicate the steady-state cell count predicted by the MSWs.

simulate drug diffusion over time using the built-in partial differential equation (PDE) solver and the PDE grid functionality. Using the PDE solver, we studied how MSWs shift with different pharmacodynamic parameters, such as the drug elimination rate $\gamma$. The parameter $\gamma$ may change depending on the drug under study, tissue type, and patient-specific drug metabolism. Example results are shown in Fig 4, with each column in the figure corresponding to a value of $\gamma$ increasing from left to right. We found that varying $\gamma$ impacts the MSW pattern (Fig 4A and 4B), with low elimination rate ($\gamma = 10^{-4}$) selecting primarily for genotype 11 (orange), while a high elimination rate ($\gamma = 0.5$) selects primarily for genotype 00 (blue). Example simulations and population timetraces are shown in Fig 4C and 4D. As shown in Fig 4D, it takes several weeks for some of the simulations to converge on the population distribution predicted by

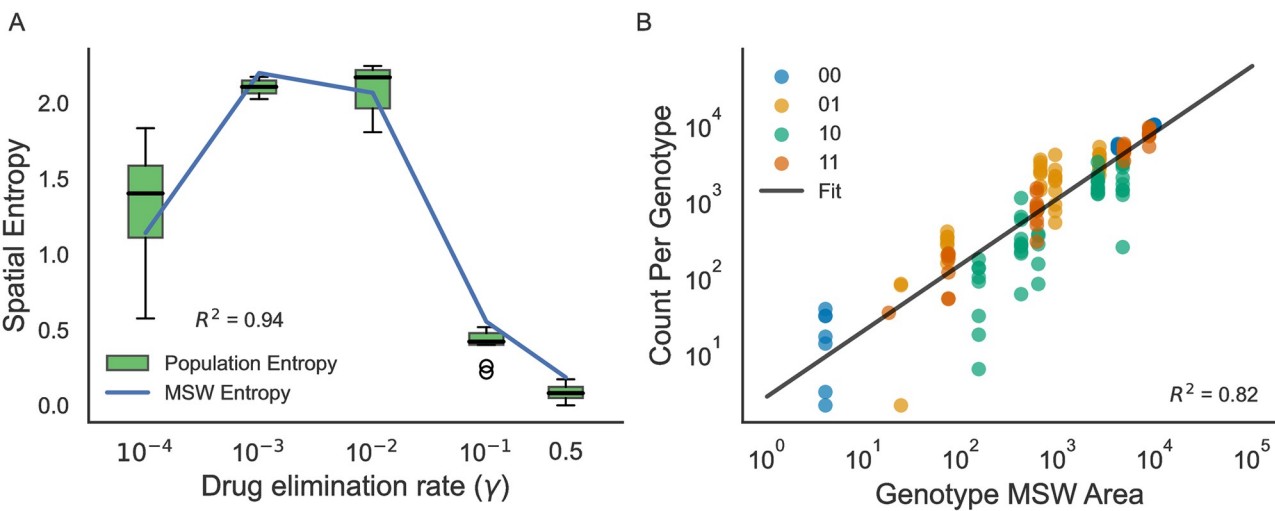

**Fig 5. MSW structure drives population heterogeneity. (A)** Altieri spatial entropy of the population genotypes and MSWs as a function of drug elimination rate. $R^2$ shown for the correlation between MSW entropy and population entropy. **(B)** Correlation between the area occupied by an MSW and the number of cells of that genotype at the end of the simulation. Each point in the scatter plot corresponds to an individual genotype in a single simulation. $N = 10$ simulations per drug elimination rate $\gamma$.

MSWs in this particular parameter regime—therefore, these simulations are most likely to be relevant for a relatively long-term treatment schedule, where the drug distribution can reach a steady-state due to repeated dosing. However, other parameter regimes and pharmacokinetic parameters may result in different rates of convergence.

To better understand the utility of MSWs in predicting evolution, we next explored how the structure of MSWs, such as spatial heterogeneity, shapes evolution. First, we investigated the impact of MSW heterogeneity on the resulting population heterogeneity. We quantified MSW and population heterogeneity using Altieri entropy, which decomposes entropy into spatial mutual information and global residual entropy [39]. We found that MSW heterogeneity is strongly associated with population heterogeneity, with $R^2 = 0.94$ (Fig 5A).

Thus far, few studies have investigated the impact of space occupied by MSWs on the evolution of drug resistance—time spent within a MSW during a treatment regimen has been the primary concern when using MSWs to design optimal therapies. We found that the space occupied by a genotype's mutant selection window is correlated with the number of cells of that genotype at the end of the simulation (Fig 5B), with $R^2 = 0.82$. Taken together, these results suggest that 1) drug diffusion can drive MSW heterogeneity, 2) MSW spatial heterogeneity shapes population spatial heterogeneity, and 3) the area occupied by a mutant selection window is important to consider when studying the evolution of drug resistance.

## Sensitivity analysis of mutation rate, initial mutant proportion, and blood vessel geometry

We hypothesized that the predictive power of MSWs would depend on certain experimental parameters such as mutation rate and initial population heterogeneity, but would not depend environmental parameters such as blood vessel geometry. To test this, we performed a sensitivity analysis by varying initial mutant proportion, mutation rate, and blood vessel separation and quantifying the difference between the Altieri entropy of the MSWs and the population.

Normalized entropy difference $\Delta_{entropy}$ was calculated as

$$\Delta_{entropy} = \frac{(e_{MSW} - e_{population})^2}{e_{MSW}}, \qquad (6)$$

where $e_{MSW}$ and $e_{population}$ refer to the Altieri entropy of the MSWs and the population, respectively. Results of the sensitivity analysis are shown in S1 Fig. We found that $\Delta_{entropy}$ decreased with increasing mutation rate up to 0.01, but beyond 0.01, drift begins to dominate over selection and the predictive power of MSWs decreases. The marginal distribution shows that initial mutant probability was not strongly correlated with $\Delta_{entropy}$. However, the joint distribution in reveals that the impact of initial mutation probability was more prominent at lower mutation rates, with increasing initial mutant probability corresponding with lower $\Delta_{entropy}$ for mutation rate less than $10^{-3}$. Notably, we found that that $\Delta_{entropy}$ did not vary strongly with vessel separation, suggesting that the predictive capacity of MSWs is independent of blood vessel density. Example simulations of varying blood vessel separation are shown in S2 Fig. These results suggest that MSWs are most likely to be relevant for predicting population structure when the mutation supply or population heterogeneity is high.

## Application to experimental non-small cell lung cancer data

The theoretical framework presented here may be applied to a variety of different settings involving spatial evolution of asexually reproducing populations. This includes both infectious diseases and solid tumors, which may demonstrate dramatically different length and time scales. For instance, a tumor may have a length scale on the order of centimeters and a treatment timescale of months to years, while a case of mitral valve endocarditis may have a length scale on the order of millimeters and a timescale of days to weeks. To better understand the broader applicability of our work, we investigated real-world examples of drug diffusion in tumors. Primeau et. al. analyzed *in vivo* perivascular doxorubicin concentration in three cell lines (murine 16C and EMT6 tumors and human prostate cancer PC-3) [40]. In all three cell lines, they found that doxorubicin concentration varied dramatically as a function of distance from blood vessels, revealing marked heterogeneity on the length scale of the tumors. Fitting their data to a 1-D model of exponential decay,

$$u(x) = ke^{-xln(2)/L}, \qquad (7)$$

where $x$ is the distance from a blood vessel and $L$ is the "characteristic penetration length" (distance at which $u(x) = 0.5 * k$), they found that $L$ is between 40–50 μm. This length scale is on the order of a typical cancer cell diameter, suggesting that different cells within the tumor experience dramatically different drug concentrations.

While most work on drug diffusion in cancer has focused on doxorubicin due to its fluorescent properties, other anti-cancer drugs have been found to also have significant gradients across the length scale of a tumor. For instance, cells 75–150 μm from the nearest blood vessel were found to regain control levels of proliferation in murine models treated with gemcitabine [41]. Other drugs, including mitoxantrone, 5-Fluorouracil, methotrexate, and danorubicin have been shown to have similar biologically relevant diffusion length scales [42, 43].

We combined the drug diffusion model in Eq (7) with *in vitro* dose-response experiments in non-small cell lung cancer (NSCLC). We engineered PC9 cells with a BRAF V600E mutation, a KRAS G12V mutation, or both, in addition to fluorescent labels. By culturing cells in different concentrations of gefitinib, an EGFR inhibitor, and imaging the cells over time, we estimated the growth rate for each cell line versus drug concentration (Fig 6A). Raw data of cell counts over time for each condition are shown in S3 Fig. Notably, we observe fitness costs

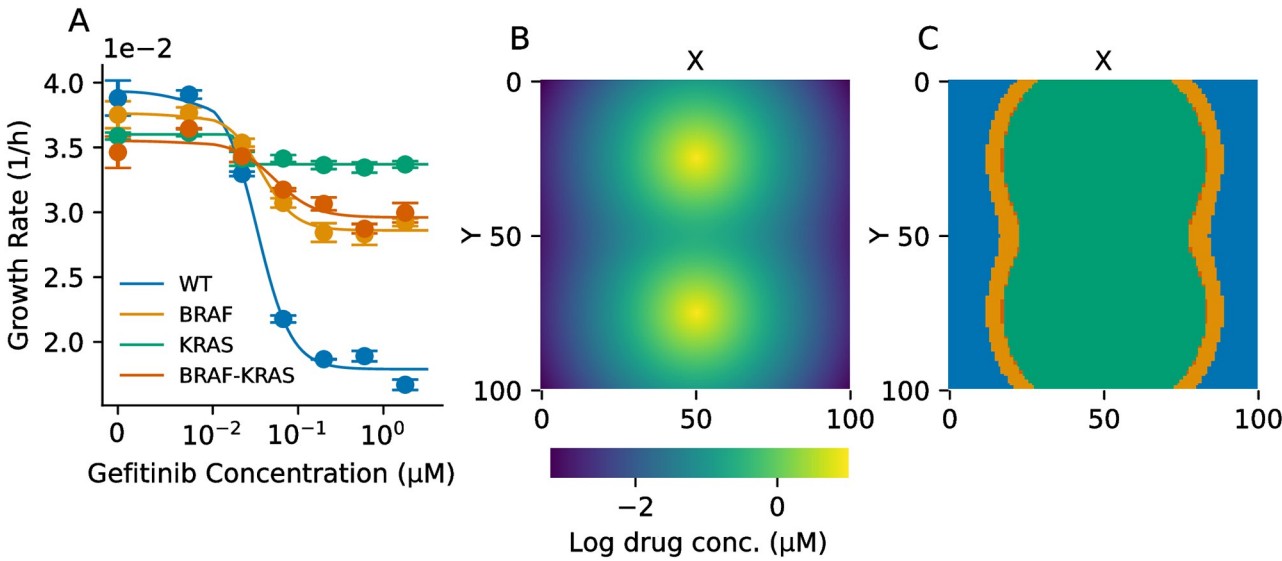

**Fig 6. Empirical fitness seascape in genetically-engineered NSCLC cells.** (A) Empirical dose-response curves for PC9 cells. WT = wild-type, BRAF = BRAF V600E mutation, KRAS = KRAS G12V mutation, BRAF-KRAS = both mutations. (B) Example drug diffusion from two blood vessels using the model in Eq (7) and characteristic length $L$ = 2 grid points ($\sim$20–40 µm). (C) MSWs introduced by the drug concentration gradient in **B** calculated using the fitness seascape in **A**.

to resistance, as all of the drug-resistant cell lines exhibit a lower growth rate than the wild-type in the absence of drug. We parameterized our agent-based model with this data and simulated drug diffusion with varying length scales using Eq (7). An example of drug diffusion with $L$ = 2 ($\sim$20–40 µm) is shown in Fig 6B and the resulting MSWs are shown in Fig 6C. We observed a steep drop off in growth rate above a gefitinib concentration of 1.8µM (S4 Fig)—thus, we defined drug concentrations above 1.8µM as the "net loss" regime, meaning that cell division probability was zero for each genotype.

We used this *in vitro* dose-response data and diffusion model to investigate how mutant selection windows impact tumor drug resistance. Using HAL, we performed stochastic, spatial simulations to model tumors subjected to a treatment regimen. The treatment plan consisted of four cycles, with one week on drug followed by one week off, followed by a subsequent four-week period without therapy. The drug regimen is indicated by gray vertical bars in Fig 7C. We quantified the size of the tumor and fraction of cells that were resistant over time for varying diffusion length scales, from $L$ = 2 to $L$ = 16 grid points (Fig 7). We found that a low characteristic length of $L$ = 2 did not fully inhibit tumor growth, but resulted in a low fraction of drug resistant cells (Fig 7C). Mutant selection windows depicted in Fig 7A indicate that the wild-type was selected across a broad area of the tumor under this characteristic length. $L$ = 4 somewhat inhibited tumor growth but resulted in tumors composed of $\sim$30% drug-resistant cells. Notably, the mutant selection windows for BRAF and KRAS mutants took up most of the tumor area in this regime. We found similar results for $L$ = 8, with the KRAS MSW occupying the most space. Finally, $L$ = 16 completely obliterated the tumor for each replicate, as the net loss regime covered the entire grid. While these results were obtained by initializing tumors with pre-existing heterogeneity (proportion of initial mutant cells $\sim$10%), we repeated these analyses with no pre-existing heterogeneity and found qualitatively similar results (S5 Fig).

These results suggest that mutant selection windows introduced by drug diffusion drives drug resistance in this model of tumor evolution. When mutant selection windows for drug

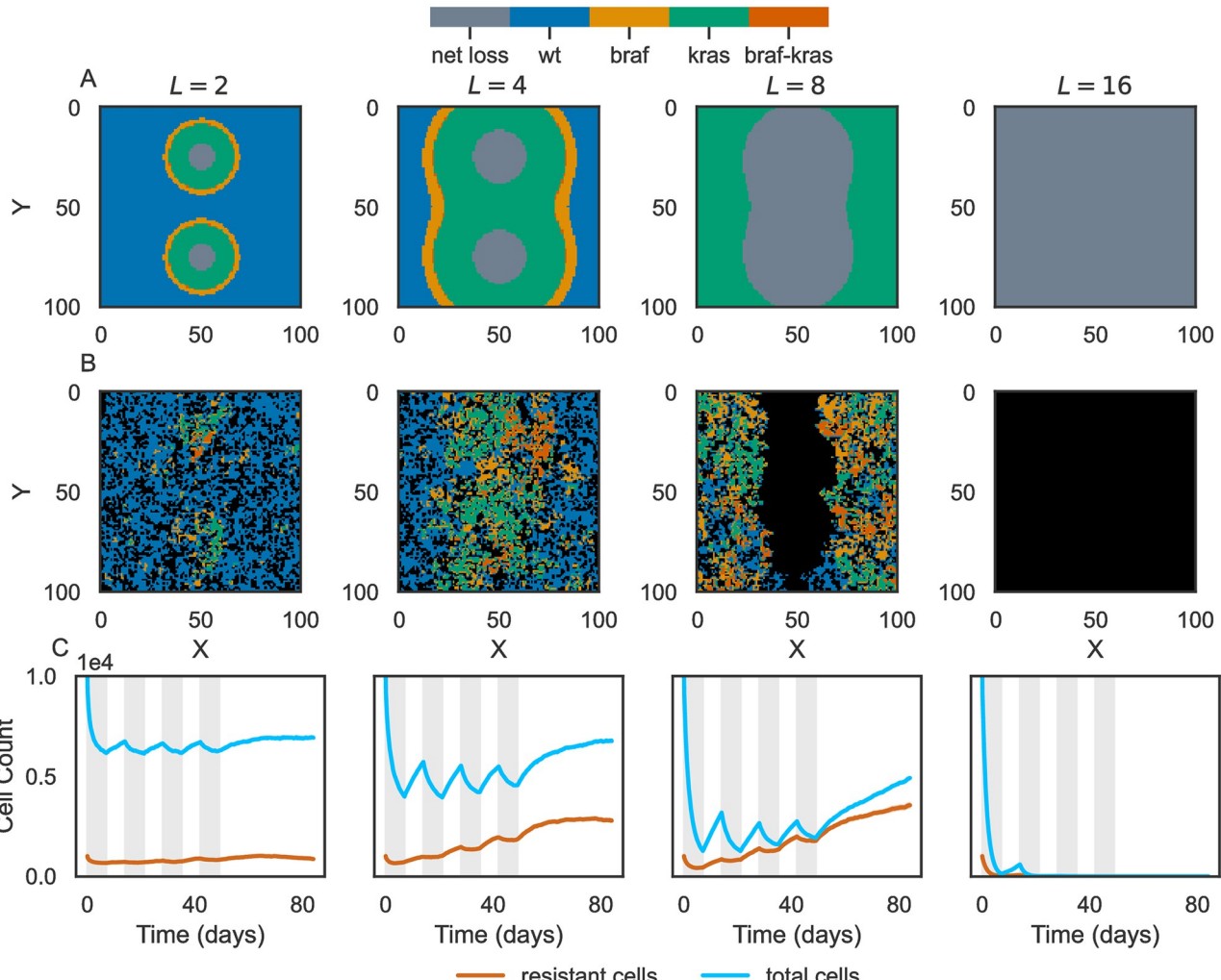

**Fig 7. Mutant selection windows drive drug resistance in simulated cancer therapy.** Summary of tumor therapy simulations. Each column corresponds to a different drug diffusion characteristic length $L$. **(A)** Mutant selection window plots for different length scales resulting from drug diffusion from two blood vessels. The net loss regime represents a drug concentration that completely inhibits cell division. **(B)** Example simulations corresponding to the characteristic length in **A**. Black grid points indicate no cells at that position. **(C)** Average timecourse of total number of cells (blue) and the number of drug resistant cells (orange). "Drug resistant" refers to any cell that is not wild-type. Traces represent the average of $N = 10$ simulations and are shaded by the standard error. In many cases, the standard error is less than the width of the plot line. Gray vertical bars indicate the "drug on" time.

resistant mutants dominate, cycled therapy fails to fully eliminate the tumor and results in a high fraction of drug resistant cells.

## Discussion

In this work, we have investigated how MSW data is embedded in fitness seascapes and how numerous MSWs may exist when the selection pressure varies in space. In a simple model of drug diffusion from a blood vessel, we have illustrated how multiple MSWs may exist simultaneously in space. Using a 2-D agent-based model of evolution, we showed how MSW structure shapes evolution of a population in a drug concentration gradient using synthetic data. These results suggest that incorporating information from fitness seascapes may expand the power of

MSW models to predict evolution. Additionally, we measured a novel empirical fitness seascape in NSCLC and studied how mutant selection windows promote tumor drug resistance *in silico*. We found that MSWs that arise due to drug diffusion may play a role in drug resistance in cancer. Importantly, our work demonstrates how certain drug diffusion regimes not only fail to inhibit tumor growth, but also promote the emergence of drug resistant mutants.

Previous computational and experimental work has demonstrated that drug diffusion gradients and differential drug penetration, which permit heterogeneous MSWs, facilitate antibiotic and antiviral resistance [6, 25, 44]. For instance, Feder et al. have explored how differential drug penetration in HIV results in variable mutant selection windows and can promote drug resistance [6]. We expand on this work to include spatial stochastic simulations, novel drug diffusion models, and a novel, empirical drug resistance model in cancer. Our work shows how the MSW model is a general framework for understanding how pharmacokinetics may promote drug resistance across diseases, including cancer and infectious disease.

These findings have several limitations. While the purpose of this work is to demonstrate how MSWs are important to consider in a variety of settings and diseases, we did not exhaustively explore the space of possible pharmacokinetic or biological parameters. Furthermore, different tissue and cell types may consume drug at variable rates, confounding the constant rate of drug elimination considered here. However, we mitigate this concern in our cancer model by relying on a drug diffusion model generated from *in vivo* data. In the context of infectious disease, pathogens may migrate along selection gradients—while we did not model this phenomenon, we expect that migration may increase the relevance of MSWs, as migration promotes admixture and increases the capacity of cells to proliferate in their "ideal" location. More complex blood vessel arrangements may also confer additional MSW heterogeneity [45]. While the results presented here pertain to drug variation in space, variation in time is also a concern. Drug pharmacokinetic profiles, dosing regimes, and patient-specific nonadherence may cause the serum drug concentration to cross multiple mutant selection windows throughout the course of treatment. For instance, Nande and Hill, among others, have shown that drug absorption rate and patient nonadherence can impact the evolution of resistance [2, 26].

While previous applications of MSWs have focused on antibiotic resistance, we believe that the concepts explored here are highly relevant to cancer, where drug resistance is a major driver of mortality. As the drug concentration profile is dependent on the tumor type, specific drug of interest, and vessel distribution, experiments accounting for these factors will be necessary for clinical translation of the theoretical concepts presented here. Previous work has demonstrated the use of radiolabeled drugs and immunofluorescence for quantifying drug diffusion *in vivo* [46]. Combining these techniques with microvasculature imaging technology such as super-resolution ultrasound imaging may enable fine-grained prediction of the drug concentration profile in an individual tumor [47].

Our work is related to several other theoretical concepts across ecology, evolution, and medicine. For instance, fitness valley crossings, which permit adaptation despite there being a substantial fitness barrier, may be more likely in spatial gradients [48]; the boundary between mutant selection windows may promote genetic admixture, facilitating fitness valley crossings [49]. Furthermore, population heterogeneity discussed here may be thought of in the context of quasispecies theory [50, 51]. Although quasispecies theory is commonly applied to viral dynamics, it may be more generally applied to any system with selection and genetic drift. A drug-dependent, spatial quasispecies theory may be comparable to the MSW model. While MSWs reflect selection only, quasispecies theory includes selection and drift and describes an equilibrium distribution of genotypes.

In the future, we may leverage fitness seascape data and the MSW analysis framework to more accurately predict the evolution of drug resistance. This may allow us to predict, or

even control the trajectory of an evolving population and leverage known mechanisms of resistance [52–54]. Such an approach could allow us to optimize drug dosing regimens, reduce the total amount of drug used in the course of treatment, and help mitigate the risk of drug resistance.

## Materials and methods

### Diffusion from a point source with constant absorption in one dimension

We modeled the drug concentration gradient that a population of cells in tissue may experience as a point source with diffusion and constant absorption. The absorption rate encapsulates clearance, metabolism, and consumption of drug. Formally, we want to track the concentration $u(x, t)$ on an infinite size domain, with diffusion described by diffusivity $D$, absorption at every position described by a rate $\gamma > 0$, and a source term that is a Kronecker delta function ($\delta(x)$) with strength $k > 0$ at the origin $x = 0$. The diffusion equation then takes the form:

$$\frac{\partial}{\partial t} u(x, t) = D \frac{\partial^2}{\partial x^2} u(x, t) - \gamma u(x, t) + k\delta(x). \tag{8}$$

We assume we initially have zero concentration everywhere, so $u(x, 0) = 0$. To solve this, we will Fourier transform the above equation:

$$u(x, t) = \frac{1}{2\pi} \int_{-\infty}^{\infty} dy \, e^{ixy} \tilde{u}(y, t), \tag{9}$$

where $\tilde{u}(y, t)$ is the Fourier transform of $u(x, t)$. Now let us take temporal and spatial derivatives of Eq (9), which we can then substitute into Eq (8):

$$\begin{aligned}
\frac{\partial}{\partial t} u(x, t) &= \frac{1}{2\pi} \int_{-\infty}^{\infty} dy \, e^{ixy} \frac{\partial}{\partial t} \tilde{u}(y, t), \\
\frac{\partial}{\partial x} u(x, t) &= \frac{1}{2\pi} \int_{-\infty}^{\infty} dy \, iy e^{ixy} \tilde{u}(y, t), \\
\frac{\partial^2}{\partial x^2} u(x, t) &= \frac{1}{2\pi} \int_{-\infty}^{\infty} dy \, (-y^2) e^{ixy} \tilde{u}(y, t).
\end{aligned} \tag{10}$$

The final fact that we need is the Fourier transform of the Dirac delta function,

$$\delta(x) = \frac{1}{2\pi} \int_{-\infty}^{\infty} dy \, e^{ixy}. \tag{11}$$

Substituting Eqs (9)–(11) into Eq (8), and collecting everything on one side under the same integral, we get:

$$\frac{1}{2\pi} \int_{-\infty}^{\infty} dy \, e^{ixy} \left( \frac{\partial}{\partial t} \tilde{u}(y, t) + Dy^2 \tilde{u}(y, t) + \gamma \tilde{u}(y, t) - k \right) = 0. \tag{12}$$

In order for Eq (12) to be true for any $x$, the parenthetical terms have to equal to zero. This yields an ordinary, first-order differential equation for $\tilde{u}(y, t)$:

$$\frac{\partial}{\partial t} \tilde{u}(y, t) = -(Dy^2 + \gamma) \tilde{u}(y, t) + k. \tag{13}$$

Using the initial condition $u(x, 0) = 0$, we find the solution for $\tilde{u}(y, t)$:

$$\tilde{u}(y, t) = \frac{-k}{Dy^2 + \gamma}\left(e^{-t(Dy^2+\gamma)} - 1\right). \tag{14}$$

Substituting into Eq (9) and rearranging we get our solution for $u(x, t)$:

$$u(x, t) = \frac{1}{2\pi}\int_{-\infty}^{\infty} dy\, \frac{e^{ixy}(1 - e^{-t(Dy^2+\gamma)})k}{Dy^2 + \gamma}. \tag{15}$$

Taking the limit $t \to \infty$, the concentration profile approaches a stationary distribution that is analytically solvable and simplifies the integration term to yield:

$$u(x, \infty) = \frac{k}{\sqrt{4D\gamma}}\exp\left(-|x|\sqrt{\frac{r}{D}}\right). \tag{16}$$

## Diffusion from a delta source with constant absorption in two dimensions

To model drug diffusion from a point source is two dimensions, we use a similar modeling technique as Eq (8):

$$\frac{\partial}{\partial t}u(x_1, x_2, t) = D\left[\frac{\partial^2}{\partial x_1^2} + \frac{\partial^2}{\partial x_2^2}\right]u(x_1, x_2, t) - \gamma u(x_1, x_2, t) + k\delta^{(2)}(x_1, x_2). \tag{17}$$

Here $\delta^{(2)}(x_1, x_2)$ is the 2D Dirac delta, $\delta^{(2)}(x_1, x_2) = \delta(x_1)\delta(x_2)$ which represents our drug source. We know by symmetry that the solution will only depend on the radial coordinate, $u(x_1, x_2, t) = u(r, t)$, where $r = \sqrt{x_1^2 + x_2^2}$. However, it is easier to proceed with the Fourier transform first in Cartesian coordinates, so we will delay the transformation to polar coordinates for now. The 2D Fourier transform is:

$$u(x_1, x_2, t) = \frac{1}{(2\pi)^2}\int_{-\infty}^{\infty} dy_1 dy_2\, e^{i(x_1 y_1 + x_2 y_2)}\tilde{u}(y_1, y_2, t). \tag{18}$$

We calculate the temporal and spatial derivatives analogously to Eqs (10) and (11), and the result is this 2D iteration of Eq (12):

$$\frac{1}{(2\pi)^2}\int_{-\infty}^{\infty} dy_1 dy_2\, e^{i(x_1 y_1 + x_2 y_2)}\left(\frac{\partial}{\partial t}\tilde{u} + D(y_1^2 + y_2^2)\tilde{u} + \gamma\tilde{u} - k\right) = 0. \tag{19}$$

We can now convert our original and Fourier variables into polar coordinates: $x_1 = r\cos\theta$, $x_2 = r\sin\theta$, $y_1 = \rho\cos\psi$, $y_2 = \rho\sin\psi$. By symmetry we know that $\tilde{u}(y_1, y_2, t) = \tilde{u}(\rho, t)$, where $\rho = \sqrt{y_1^2 + y_2^2}$. Eq (19) becomes:

$$\frac{1}{(2\pi)^2}\int_0^{\infty} d\rho \int_0^{2\pi} d\psi\, \rho\, e^{ir\rho\,\cos(\psi-\theta)}\left(\frac{\partial}{\partial t}\tilde{u}(\rho, t) + D\rho^2\tilde{u}(\rho, t) + \gamma\tilde{u}(\rho, t) - k\right) = 0. \tag{20}$$

For this equation to be always true, the parenthetical terms must be zero, yielding:

$$\frac{\partial}{\partial t}\tilde{u}(\rho, t) = -(D\rho^2 + \gamma)\tilde{u}(y, t) + k. \tag{21}$$

This has exactly the same form as Eq (13) with $\rho$ in place of $y$. Hence the solution mirrors Eq

(14):

$$\tilde{u}(\rho, t) = \frac{-k}{D\rho^2 + \gamma}\left(e^{-t(D\rho^2 + \gamma)} - 1\right).$$

(22)

The solution $u(x_1, x_2, t) = u(r, \theta, t)$ in polar coordinates is then the inverse Fourier transform of the above:

$$
\begin{aligned}
u(r, \theta, t) &= \frac{1}{(2\pi)^2}\int_0^\infty d\rho \int_0^{2\pi} d\psi\, \rho\, e^{ir\rho\cos(\psi-\theta)} \frac{k}{D\rho^2 + \gamma}\left(1 - e^{-t(D\rho^2 + \gamma)}\right) \\
&= \frac{1}{2\pi}\int_0^\infty d\rho\, \frac{k\rho J_0(r\rho)}{D\rho^2 + \gamma}\left(1 - e^{-t(D\rho^2 + \gamma)}\right).
\end{aligned}
$$

(23)

Here $J_0(z)$ is a Bessel function of the first kind. Note that the $\theta$ dependence has been integrated out, so $u(r, \theta, t) = u(r, t)$, as expected by symmetry. This integral cannot be computed analytically, but can be easily approximated numerically. To solve for the steady-state solution, we take the limit $t \to \infty$:

$$
\begin{aligned}
u(r, t = \infty) &= \frac{1}{2\pi}\int_0^\infty d\rho\, \frac{k\rho J_0(r\rho)}{D\rho^2 + \gamma} \\
&= \frac{k}{2\pi D} K_0\left(\sqrt{\frac{\gamma}{D}}r\right).
\end{aligned}
$$

(24)

Here $K_0(z)$ is a modified Bessel function of the second kind. This function diverges at $r = 0$, which proves problematic for calculating concentrations near the blood vessel sources. Instead, we choose a blood vessel radius $r > 0$ and set the drug concentration within that radius equal to a constant maximum drug concentration. Thus, this analysis holds more strongly for analysis away from the immediate vicinity of the vessel.

To model drug diffusion from an arbitrary number of sources, we convolved the discretized version of Eq (24) ($u_{i,j}(t = \infty)$) with a 2D matrix of point sources ($\Delta_{i,j}$), where $(i, j)$ represents position in the discretized 2D space:

$$U_{i,j} = u_{i,j}(t = \infty)\circledast\Delta_{i,j}.$$

(25)

## Evolutionary simulations with HAL

We used HAL to implement on-lattice 2-dimensional agent-based simulations with drug diffusion. Each cell, or agent, was defined by its genotype, modeled by the binary strings 00, 01, 10, and 11, with each position in the string corresponding to resistance-conferring point mutation. These genotypes were assigned to the synthetic or empirical dose response curves shown in Figs 1A and 6A. These dose-response curves determined the division probability of each cell as a function of drug concentration. At each time step, cells have the opportunity to divide with or without mutation, die, or do nothing. To divide, a cell must have at least one adjacent grid space unoccupied. When mutating, cells can change genotype to any of the two "adjacent" genotypes, meaning genotypes which differ by 1 position in the binary string (i.e., genotype 10 can mutate to 11 or 00 with equal probability). A lattice of size 100x100 was used for all simulations. For the simulations used in Figs 4 and 5, blood vessels were placed at positions $x = 50$, $y = 25$ and $x = 50$, $y = 75$. Diffusion was modeled using the PDE solver functionality in HAL, with the blood vessels supplying a constant drug concentration. Each point in the lattice

**Table 1. Simulation parameters.** Synthetic simulation refers to the results in Figs 4 and 5, while empirical simulation refers to the results in Figs 6 and 7.

| Parameter | Synthetic Data Simulation Value | Empirical Data Simulation Value |
|---|---|---|
| Mutation rate | $10^{-4}$ $bp^{-1}$ | $10^{-3}$ $bp^{-1}$ |
| Initial mutant probability | 0.01 | 0.01 |
| Death rate | 0.1 | 0.025 |
| Initial grid density | 1 | 0.01–1 |
| Initial shape | Circle ($r = 10$) | Square ($L = 100$) |
| Blood vessel drug concentration | $10^3$ µg/mL | 10 µg/mL |
| Diffusion rate | 0.1 | n/a |
| Time steps | 1000 hr | 2016 hr |

absorbed drug at a variable rate $\gamma$, and the boundary was set to drug concentration of 0. For the constant drug concentration profile used in the NSCLC experiments, the drug concentration at each lattice point was calculated according to the pharmacokinetic model in Eq (7) prior to the start of the simulation. Cells were subject to the constant drug concentration profile during the "drug on" time and 0 drug during the "drug off" time. Parameters used in the simulations for Figs 4–7 are shown in Table 1.

For the sensitivity analyses, mutation rate and initial mutant probability were varied between 0 and $10^{-1}$ and blood vessel spatial separation was varied from 0 to 80 grid points (S1 Fig).

## Dose-response curves in genetically engineered cells lines

To parameterize our simulations, we performed an in-vitro drug sensitivity profiling assay using parental PC-9 cells (Sigma-Aldrich, USA) expressing green fluorescence protein (GFP). Drug resistant PC-9 cells were engineered using a lentiviral transduction system to stably integrate and co-express B-RAF-V600E and mCherry, K-RAS-G12V and mCherry, and double mutant expressing both oncogenes and blue fluorescence protein (BFP). Cells were maintained in RPMI-1640 supplemented with 10% heat-inactivated Fetal Bovine Serum (FBS) and 1% penicillin and streptomycin at 37 C under a humidified atmosphere containing 5% CO2. Experimentally, cells were harvested at 70–80% confluence, stained with trypan blue (Invitrogen, USA), and counted with a Countess 3 Automated Cell Counter (Life Technologies, USA). Each cell line was seeded at a density of 1500 cells/100µL in a 96-well plate (Corning, USA) that contains different concentrations of Gefitinib (Cayman, USA) ranging from 0–5.4 µM in three replicates per condition. Then time-lapse microscopy images were obtained for GFP, mCherry, and BFP using BioSpa (Biotek, USA) every 4 hours for 96 hours. Images were processed with the open-source software CellProfiler to obtain cell counts. The CellProfiler pipeline is available in the github repository.

Growth rates for each condition were estimated by fitting a linear regression to the log-transformed cell counts over time. Growth rate versus drug concentration were then fit to the following dose-response curve using the optimize package in SciPy [55]:

$$g([c]) = gmax + \frac{(g_{min} - g_{max}) * [c]^{\nu}}{IC_{50}^{\nu} + [c]^{\nu}},$$

(26)

where $[c]$ is the drug concentration, $g_{max}$ is the maximum (drug-free) growth rate, $g_{min}$ is the minimum growth rate, $IC_{50}$ is the half-maximal inhibitory drug concentration, and $\nu$ is the Hill coefficient. Estimated dose-response parameters are shown in Table 2.

**Table 2. Estimated dose-response parameters.**

| Parameter | WT | BRAF | KRAS | BRAF-KRAS |
|---|---|---|---|---|
| gmax (hr$^{-1}$) | 0.039 | 0.037 | 0.036 | 0.035 |
| gmin (hr$^{-1}$) | 0.018 | 0.029 | 0.033 | 0.029 |
| IC$_{50}$ (µg/mL) | 0.033 | 0.037 | 0.020 | 0.053 |
| $\nu$ | 2.18 | 2.19 | 17.45 | 1.85 |

## Supporting information

**S1 Fig. Sensitivity analysis reveals model robustness to vessel geometry. (A)** Joint heatmap of mutation rate versus initial mutant probability. Normalized entropy difference is the squared difference between the Altieri entropy of the MSW map and the Altieri entropy of the final population distribution, normalized by the MSW entropy (Eq (6)). **(B)** Joint heatmap of vessel separation (lattice points between the two blood vessels) versus initial mutant probability. **(C)** Marginal distribution of normalized entropy difference versus blood vessel separation. **(D)** Marginal distribution of normalized entropy difference versus initial mutant probability. **(E)** Marginal distribution of normalized entropy difference versus mutation rate.
(TIF)

**S2 Fig. Example MSWs and population distributions for vessel separation experiments.** Each column corresponds to a different blood vessel separation distance labeled at the top of each column. Distance is in units of lattice points. Drug elimination rate $\gamma = 0.01$, mutation rate = 0.001, and initial mutant probability = 0.01.
(TIF)

**S3 Fig. Cell count over time for different concentrations of gefitinib.** Each column corresponds to a cell type (WT, BRAF, KRAS, or BRAF-KRAS). Each row corresponds to a concentration of gefitinib (labeled on the right hand side in µM). Each condition has 3 replicates. Each condition is labeled with the estimated average growth rate (hr$^{-1}$).
(TIF)

**S4 Fig. Dose-response curves with corresponding pharmacodynamic curve fits used to parameterize tumor agent-based models.** Growth rate versus drug concentration calculated from the data in S3 Fig.
(TIF)

**S5 Fig. Mutant selection windows drive drug resistance in simulated cancer therapy.** Summary of tumor therapy simulations with no pre-existing heterogeneity. Each column corresponds to a different drug diffusion characteristic length $L$. **(A)** Mutant selection window plots for different length scales resulting from drug diffusion from two blood vessels. The net loss regime represents a drug concentration that completely inhibits cell division. **(B)** Example simulations corresponding to the characteristic length in **A**. Black grid points indicate no cells at that position. **(C)** Average timecourse of total number of cells (blue) and the number of drug resistant cells (orange). Drug resistant refers to any cell that is not wild-type. Traces represent the average of $N = 10$ simulations and are shaded by the standard error. In many cases, the standard error is less than the width of the plot line. Gray vertical bars indicate the "drug on" time.
(TIF)

## Author Contributions

**Conceptualization:** Eshan S. King, Beck Pierce, Michael Hinczewski, Jacob G. Scott.

**Formal analysis:** Eshan S. King, Beck Pierce, Michael Hinczewski.

**Investigation:** Eshan S. King, Dagim S. Tadele, Beck Pierce.

**Methodology:** Eshan S. King, Dagim S. Tadele.

**Software:** Eshan S. King, Beck Pierce.

**Supervision:** Michael Hinczewski, Jacob G. Scott.

**Visualization:** Eshan S. King, Beck Pierce.

**Writing – original draft:** Eshan S. King, Beck Pierce.

**Writing – review & editing:** Dagim S. Tadele, Beck Pierce, Michael Hinczewski, Jacob G. Scott.

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
