## [Decision Letter · Decision Letter 0]

21 Jun 2023

Dear Mr. King,

Thank you very much for submitting your manuscript "Fitness seascapes promote genetic heterogeneity through spatiotemporally distinct mutant selection windows" for consideration at PLOS Computational Biology.

As with all papers reviewed by the journal, your manuscript was reviewed by members of the editorial board and by several independent reviewers. In light of the reviews (below this email), we would like to invite the resubmission of a significantly-revised version that takes into account the reviewers' comments. Three reviews have been obtained, and the recommendations range from reject to minor revisions, so it will be important to address all concerns raised by reviewers in detail. 

We cannot make any decision about publication until we have seen the revised manuscript and your response to the reviewers' comments. Your revised manuscript is also likely to be sent to reviewers for further evaluation.

Sincerely,

Dominik Wodarz

Academic Editor

PLOS Computational Biology

Natalia Komarova

Section Editor

PLOS Computational Biology

Reviewer's Responses to Questions

**Comments to the Authors:**

Reviewer #1: This work has definitely merit, as it shows the effects that spatially and temporally varying drug concentration has on genotype heterogeneity and, more specifically, on the emergence of mutant drug-resistant genotypes. However:

1) No particular infectious disease has been considered as an application.

2) The implications of the findings on drug administration treatment strategies have not been discussed.

3) Also, while the effects of the spatial distribution of the drug concentration are important, and this is indeed an important novelty aspect of the present work, the results shown in Figure 2 are expected. Thus, I would suggest to incorporate an additional example, with a more complex blood vessel topography, that preferably comes from a certain infection; this would strengthen the computational aspect of the work as well as the usefulness of the model in treatment strategies.

I think that by addressing the points above, the impact and relevance of the work will be increased significantly.

Last, while it is stated that this work is on infectious diseases, I would like to see a comment relating this model (or making a suggestion) to heterogeneity in tumors after treatment.

Reviewer #2: Comments attached.

Reviewer #3: review in attachment

**Have the authors made all data and (if applicable) computational code underlying the findings in their manuscript fully available?**

Reviewer #1: Yes

Reviewer #2: Yes

Reviewer #3: Yes

PLOS authors have the option to publish the peer review history of their article (what does this mean?). If published, this will include your full peer review and any attached files.

Reviewer #1: No

Reviewer #2: No

Reviewer #3: No
---

## [Decision Letter · Decision Letter 1]

11 Oct 2023

Dear Mr. King,

Thank you very much for submitting your manuscript "Diverse mutant selection windows shape spatial heterogeneity in evolving populations" for consideration at PLOS Computational Biology.

As with all papers reviewed by the journal, your manuscript was reviewed by members of the editorial board and by several independent reviewers. In light of the reviews (below this email), we would like to invite the resubmission of a significantly-revised version that takes into account the reviewers' comments.

Reviewers appreciated the additional components that have been added to the manuscript in response to the review. At the same time, they felt that those additional components could be refined in a further revision. Please revise the paper thoroughly with all reviewer comments in mind. 

We cannot make any decision about publication until we have seen the revised manuscript and your response to the reviewers' comments. Your revised manuscript is also likely to be sent to reviewers for further evaluation.

Sincerely,

Dominik Wodarz

Academic Editor

PLOS Computational Biology

Natalia Komarova

Section Editor

PLOS Computational Biology

Reviewer's Responses to Questions

**Comments to the Authors:**

Reviewer #1: I remain of the opinion that, in order for the applicability of this work to be demonstrated, the authors should incorporate an example of a particular disease/treatment, or at least provide rigorous connections between their examples (and the parameter values used therein) and particular infection cases encountered in real life.

Reviewer #2: I made extensive comments during the first round of review. Many of these comments still hold, so I will keep this set of comments brief. Of particular importance is a more direct discussion of the relevant length and timescales on which the discussed phenomenology are relevant. Estimates of length and timescales for the drugs and pathogens/tumors of interest and a comparison to the parameters used in the model are crucial for any downstream applicability and broader interest/relevance.

The authors have made some changes to the manuscript, including adding agent-based simulations. However, these simulations are not described in sufficient detail, nor do they measure the relevant case of de novo evolution of resistance mutants. Rather, they simulate selection on standing variation ("an initial proportion of mutants of 0.1"). This latter scenario may have some relevance, but is much less likely, given that resistance is often met with some sort of growth tradeoff, as has been measured, for instance in Pinheiro, et al. Nat Ecol Evol (2021).

Moreover, the simulations and the subsequent PDE analysis miss key physical processes -- namely that drugs are typically differentially taken up by pathogen vs host cells. Tetracyclines, for instance, are a well-known example of this. When this is the case, the drug concentration field will depend on and evolve with the population of pathogen cells (e.g. the single absorption rate assumption is unlikely to be valid). Therefore drug concentrations typically will not have a steady state concentration profile, and dynamics must be analyzed.

Reviewer #3: The revised manuscript is greatly improved from the original submission. I applaud the authors' inclusion of agent-based simulations, which have greatly strengthened the result and added more depth to the analysis. The authors have adequately addressed my previous concerns with the manuscript. However, given the inclusion of additional work (i.e. stochastic simulations and their comparison with the MSW) and the shift of focus of the manuscript, I have a number of additional issues which I believe should be addressed before publication.

1. With regards to the spatial agent-based simulations: if I understand correctly, simulations are initiated with a sparsely populated lattice (line 116-117: "Each simulation was initiated with a random sparsely populated lattice (initial density = 0.01 cells per grid position, on average)"), and then allowed to grow to confluency. How realistic is this situation with regards to the systems of interest? For example in cancer, one would expect the the population to already be at "confluency" (i.e. some sort of tissue capacity) when the drug is introduced. In the case of bacteria this may be more realistic, however bacterial cells will generally have some capacity for self-driven motility which likely complicates the picture of selection, as self-assembly and pattern formations can occur (see e.g. 10.1103/PhysRevLett.100.218103 or 10.1073/pnas.1001994107), and cells would be able to move to their ideal position in the gradient. Have you explored how different initial conditions in the population density affect the time evolutions?

2. 130-132: "The difference between the MSW entropy and mean population entropy at γ = 10^−4 is likely due to the fact that the evolving population has not reached steady-state by the end of the simulation, as indicated by the changing cell counts in Fig. 3D." Could this not be tested by running simulations for a longer time, until they reach steady-state? Is there a particular reason simulations were ended at 500h? Given that the MSW prediction for selected genotypes was calculated using a steady state assumption, would it not make sense to compare them with the simulations only when steady state is reached?

3. The time dynamics shown in Figure 3D provide great additional insight into the agent-based simulations. It would be interesting to show the MSW population size predictions as additional horizontal lines in the same plots, to see whether the agent based simulations indeed evolve toward the MSW-predicted steady-states.

4. In the section "Sensitivity analysis of mutation rate, initial mutant probability, and blood vessel separation", why introduce "mutation supply = mutation_rate x max_population_size" as a new parameter, if the max population size is kept fixed? How is this different from studying variation of the mutation rate? It would be clearer to simply use the mutation rate here as parameter of interest.

5. In Figure S1, it would be nice to also see the marginal distribution of the entropy difference for the mutation supply (or mutation rate). I.e. a similar plot as S1C and D, but for the mutation supply.

**Have the authors made all data and (if applicable) computational code underlying the findings in their manuscript fully available?**

Reviewer #1: Yes

Reviewer #2: Yes

Reviewer #3: Yes

PLOS authors have the option to publish the peer review history of their article (what does this mean?). If published, this will include your full peer review and any attached files.

Reviewer #1: No

Reviewer #2: No

Reviewer #3: No
---

## [Decision Letter · Decision Letter 2]

31 Jan 2024

Dear Dr. Scott,

We are pleased to inform you that your manuscript 'Diverse mutant selection windows shape spatial heterogeneity in evolving populations' has been provisionally accepted for publication in PLOS Computational Biology.

Also, please check a minor question that one of the reviewer poses, to avoid potential confusion.

Best regards,

Dominik Wodarz

Academic Editor

PLOS Computational Biology

Natalia Komarova

Section Editor

PLOS Computational Biology

Reviewer's Responses to Questions

**Comments to the Authors:**

Reviewer #1: I think that the points I have raised in previous revision rounds have been adequately addressed; thus, I recommend the publication of this work.

Reviewer #3: The revised manuscript includes an experimental parameterization of the model through in vitro culturing of a non-small cell lung cancer cell line with genetically induced resistant mutations. This addition adds a tangible link with the biology and nicely illustrates the real-world applicability of the dose-response curves. I have no comments on this section, and all of my previous comments have been adequately addressed.

However, related to my first comment, one small point of confusion remains in the text currently. In their response the authors note:

"Based on this feedback, we initialized all simulations as confluent grids, meaning

that every grid position is occupied by a cell."

From Figure 4D I can see that indeed the wild type now initiates at maximal confluency. However, line 122 states: "Each simulation was initiated as a circle with radius 10", which I believe is how the system was implemented in the previous version of the manuscript. I believe this no longer accurately describes the simulation?

**Have the authors made all data and (if applicable) computational code underlying the findings in their manuscript fully available?**

Reviewer #1: Yes

Reviewer #3: Yes

PLOS authors have the option to publish the peer review history of their article (what does this mean?). If published, this will include your full peer review and any attached files.

Reviewer #1: No

Reviewer #3: No

---

## [Editor Report · Acceptance letter]

16 Feb 2024

PCOMPBIOL-D-23-00565R2 

Diverse mutant selection windows shape spatial heterogeneity in evolving populations

Dear Dr Scott,

I am pleased to inform you that your manuscript has been formally accepted for publication in PLOS Computational Biology. Your manuscript is now with our production department and you will be notified of the publication date in due course.

With kind regards,

Zsofi Zombor
